# Enterovirus A71 crosses a human blood–brain barrier model through infected immune cells

Léa Gaume,[1] Hélène Chabrolles,[1,2] Maxime Bisseux,[1,2] Igor Lopez-Coqueiro,[1] Lucie Dehouck,[3] Audrey Mirand,[1,2] Cécile Henquell,[1,2] Fabien Gosselet,[3] Christine Archimbaud,[1,2] Jean-Luc Bailly[1]

**ABSTRACT**  Enterovirus A71 (EV-A71) is associated with neurological conditions such as acute meningitis and encephalitis. The virus is detected in the bloodstream, and high blood viral loads are associated with central nervous system (CNS) manifestations. We used an *in vitro* blood–brain barrier (BBB) model made up of human brain-like endothelial cells (hBLECs) and brain pericytes grown in transwell systems to investigate whether three genetically distinct EV-A71 strains (subgenogroups C1, C1-like, and C4) can cross the human BBB. EV-A71 poorly replicated in hBLECs, which released moderate amounts of infectious viruses from their luminal side and trace amounts of infectious viruses from their basolateral side. The barrier properties of hBLECs were not impaired by EV-A71 infection. We investigated the passage through hBLECs of EV-A71-infected white blood cells. EV-A71 strains efficiently replicated in immune cells, including monocytes, neutrophils, and NK/T cells. Attachment to hBLECs of immune cells infected with the C1-like virus was higher than attachment of cells infected with C1-06. EV-A71 infection did not impair the transmigration of immune cells through hBLECs. Overall, EV-A71 targets different white blood cell populations that have the potential to be used as a Trojan horse to cross hBLECs more efficiently than cell-free EV-A71 particles.

**IMPORTANCE**  Enterovirus A71 (EV-A71) was first reported in the USA, and numerous outbreaks have since occurred in Asia and Europe. EV-A71 re-emerged as a new multirecombinant strain in 2015 in Europe and is now widespread. The virus causes hand-foot-and-mouth disease in young children and is involved in nervous system infections. How the virus spreads to the nervous system is unclear. We investigated whether white blood cells could be infected by EV-A71 and transmit it across human endothelial cells mimicking the blood–brain barrier protecting the brain from adverse effects. We found that endothelial cells provide a strong roadblock to prevent the passage of free virus particles but allow the migration of infected immune cells, including monocytes, neutrophils, and NK/T cells. Our data are consistent with the potential role of immune cells in the pathogenesis of EV-A71 infections by spreading the virus in the blood and across the human blood–brain barrier.

**KEYWORDS**  neuroinvasion, EV-A71 neurotropism, blood–brain barrier model, BBB crossing, white blood cells

S ymptomatic enterovirus A71 (EV-A71) infections affect mainly children under 5 years and are more severe when they occur during the period from birth to 3 years of age (1). EV-A71 causes mainly uncomplicated medical conditions and less frequently severe neurological diseases (0.1%–1.1% cases), which can evolve to death in 0.01%–0.03% of cases (2). The most frequent, self-limiting pediatric conditions are hand-foot-and-mouth disease (HFMD)—a vesicular rash on the hands, feet, mouth, or buttocks—and herpangina, which manifests as ulcerations in the oral cavity. Severe EV-A71 infection cases

Address correspondence to Jean-Luc Bailly, j-luc.bailly@uca.fr.

The authors declare no conflict of interest.

See the funding table on p. 13.

are characterized by neurological manifestations such as headache, acute meningitis, and brainstem encephalitis (3). In young patients, the disease may progress to cardiopulmonary failure or pulmonary edema after the onset of central nervous system (CNS) involvement (4). Pathological investigations of EV-A71 fatal cases show that viral antigens are present in the cytoplasm of neurons, in particular within the brainstem, diencephalon, basal ganglia, cerebellar dentate nucleus, and the anterior horn cells of the spinal cord (5). EV-A71 is not detected in all injured neurons, and the viral antigens are focally distributed in one or a few adjacent neurons and are sometimes not detected in the most intensely inflamed areas (5). These findings suggest that active viral infection is not the only cause of lesions found in the CNS and that inflammatory changes are also involved.

As a member of the *Enterovirus* genus (Picornaviridae family), EV-A71 is one of 25 genetically related types that are assigned to the EV-A species. EV-A71 strains are assigned to seven genogroups designated A to G (6). Since the 1970s, all viruses reported in outbreaks have been assigned to genogroups B and C and notably in China to subgenogroup C4 (7). In Europe, the most prevalent strains have been assigned to subgenogroups C1 and C2 (8). A genetically divergent strain within subgenogroup C1, reported in 2015 in Germany, was associated with an outbreak of brainstem encephalitis in 2016 in Spain (9, 10). This C1-like virus has a multirecombinant origin and was associated with neurological manifestations in France (11). This virus retained only the structural protein gene sequences from the C1 viruses circulating before its emergence and inherited the other genomic sequences through genetic recombination with other types of the EV-A species.

EV-A71 is transmitted mainly via the oral route and is excreted in the feces of infected subjects. The virus generates high viral loads in the throat and small intestine and is present in the associated lymphoid tissues, the tonsils, and Peyer's patches (12–14). EV-A71 is also detected in the bloodstream, but studies investigating viremia in patients with laboratory-confirmed infection are scarce (15, 16). In an investigation of patients with EV-A71 infection, Cheng *et al.* (16) reported that blood viral loads were moderate (3.46 $\log_{10}$ RNA copies/mL) and that the median viral load (3.68 $\log_{10}$ RNA copies/mL) during days 1 to 3 after disease onset was significantly higher than the viral load (3.13 $\log_{10}$ RNA copies/mL) on day 4 and later (16). In addition, viremia detected beyond 3 days after disease onset correlated with more severe disease. In mouse and monkey models of EV-A71 infection, high viral loads in blood were associated with CNS manifestations (17, 18). EV-A71 is a neurotropic virus, and multiple routes have been proposed for neuroinvasion including retrograde axonal transport, crossing of the blood–brain barrier (BBB), and transport within infected immune cells through the BBB, known as the Trojan horse mechanism (4, 5). Retrograde axonal transport is considered the main route for CNS infection by EV-A71 (19). Because of the presence of the virus in the blood, the BBB is a possible route for virus access to the CNS.

Earlier, we used the brain microvascular endothelial cell line hCMEC/D3 to investigate the interaction of a large array of EV types, including EV-A71, with an *in vitro* human BBB model (20). Although the hCMEC/D3 cells have characteristic endothelial markers and allow the production of a reproducible BBB model, a number of limitations are associated with this immortalized cell line, such as gaps in the expression of certain transporters and metabolic enzymes compared with *in vivo* conditions (21). A more appropriate *in vitro* model of the human BBB was developed with primary endothelial cells (ECs) derived from CD34[+] hematopoietic progenitor cells differentiated in the presence of brain pericytes (22, 23). This cellular model—hereafter referred to as human brain-like ECs or hBLECs—was used to investigate the neuroinvasion process of severe acute respiratory syndrome coronavirus 2 (SARS-CoV-2) and Zika virus (24, 25). In the present study, we used the hBLEC model to investigate whether cell-free infectious EV-A71 particles or EV-A71-infected immune cells can cross the human BBB and report our findings on three EV-A71 isolates phylogenetically assigned to subgenogroups C1, C1-like, and C4, which represent distinct clades of the circulating virus.

## RESULTS

### Barrier properties of hBLECs infected with EV-A71

First, we explored the susceptibility to EV-A71 of undifferentiated CD34$^+$-derived ECs, which have features of peripheral ECs (22), and found that they were moderately susceptible (see Supplemental Material; Fig. S1). Compared with the highly susceptible rhabdomyosarcoma (RD) cells, the proportions of CD34$^+$-ECs infected with EV-A71 were 5- to 13.2-fold lower than those of infected RD cells (Fig. S2). The susceptibility to EV-A71 of brain pericytes was also tested: these cells did not support EV-A71 RNA replication (data not shown). We then investigated the susceptibility to EV-A71 of fully differentiated hBLECs. The barrier properties of mock-infected (MI) hBLECs were monitored 7, 11, and 14 days after initiation of the coculture with pericytes (Fig. S3). Our findings on the permeability to Lucifer yellow (LY, a compound known to poorly cross the intact BBB), accumulation of rhodamine 123 (a substrate of the P-glycoprotein efflux pump), and expression of zonula occludens (ZO-1) and claudin-5 (CLDN-5) proteins (Fig. S3 to S5) were consistent with previously published data on this BBB model (22, 23). hBLECs were inoculated (multiplicity of infection, MOI = 1) from the luminal side 11 days after initiation of the coculture, and permeability to LY was monitored at 24, 48, and 72 hours post-infection (hpi) (Fig. 1). The mean permeability of EV-A71-infected hBLECs was not different from that of MI hBLECs (mean, $0.58 \times 10^{-3}$ cm.min$^{-1}$; range, $0.45$–$0.7 \times 10^{-3}$ cm.min$^{-1}$) except for the C1-06 virus (mean, $0.90 \times 10^{-3}$ cm.min$^{-1}$; range, $0.79$–$1 \times 10^{-3}$ cm.min$^{-1}$). However, the permeability of C1-06-infected hBLECs was below the threshold, defining a robust *in vitro* BBB model (26). This pattern was unchanged at 48 hpi (data not shown) and 72 hpi (mean, $0.75 \times 10^{-3}$ cm.min$^{-1}$; range, $0.47$–$1.1 \times 10^{-3}$ cm.min$^{-1}$). In addition to EV-A71 strains, we tested an echovirus 6 strain, hereafter designated Echo-6. We included this virus as a cytolytic control of ECs, known to replicate actively in hCMEC/D3 cells and disrupt an *in vitro* BBB (20). For hBLECs infected with the Echo-6 strain (Fig. 1A), the permeability increased at 24 hpi (mean, $1.34 \times 10^{-3}$ cm.min$^{-1}$; range, $1.07$–$1.66 \times 10^{-3}$ cm.min$^{-1}$; *P*-value < 0.01) and 48 hpi (mean, $2 \times 10^{-3}$ cm.min$^{-1}$; range, $1.54$–$2.77 \times 10^{-3}$ cm.min$^{-1}$; data not shown). At 72 hpi, we observed an extensive cytopathic effect.

The infected hBLECs were analyzed at 24 hpi by confocal microscopy after immunofluorescence (IF) staining to detect viral dsRNA, and for each virus strain, the number of infected cells was determined after imaging the hBLEC monolayers (Fig. 1B). There was little difference between the three EV-A71 strains in the percentages of hBLECs showing viral dsRNA. The lowest percentage of infected cells was determined for C4 (mean, 0.4%, range 0%–1.4%). For the other two EV-A71 strains, the mean percentage of infected hBLECs ranged between 2.1% (C1-06) and 2.7% (C1-16). With 13% of infected cells (range, 6.3%–20%), Echo-6 showed differences compared with EV-A71. No impact of EV-A71 infection was observed on tight junctions (TJs) (Fig. S4) and efflux pumps (Fig. S5). To summarize, the permeability to LY of EV-A71-infected hBLECs remained low, suggesting a limited impact of virus replication on the barrier properties *in vitro*, a pattern consistent with the low proportion of infected hBLECs at early timepoints.

### Replication kinetics of EV-A71 in hBLECs

We examined whether hBLECs could produce infectious viruses while their barrier properties were not affected by EV-A71. The kinetics of EV-A71 replication in hBLECs were monitored by semi-quantitative RT-qPCR at the indicated timepoints pi by detecting the total viral RNA genomes released in the upper and lower compartments from the luminal and basolateral sides of hBLECs, respectively (Fig. 2A through C). In hBLECs infected with the C4 strain, the viral load in the upper compartment increased from 6 to 48 hpi, as indicated by the decrease in the mean Ct values from 22.09 to 16.10 (Fig. 2A). Compared with the pattern determined in the upper compartment, the release of viral RNAs in the lower compartment had a reverse trend, and the amounts of viral RNAs determined at all timepoints pi were lower. The release of viral RNAs in the upper and

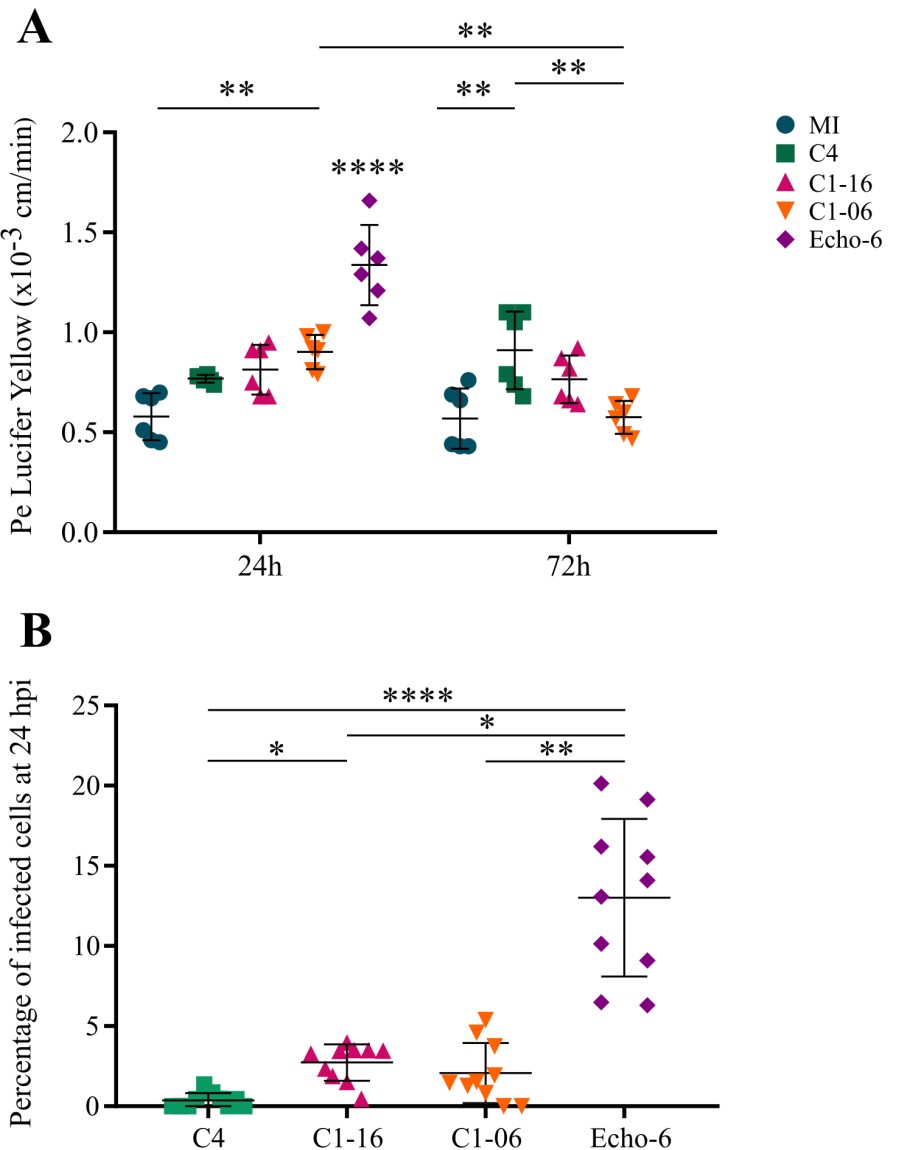

**FIG 1** Permeability of human brain-like endothelial cells (hBLECs) after infection of three enteroviruses A71 (EV-A71). (A) Permeability coefficient (Pe) of Lucifer yellow determined after inoculation of hBLECs (MOI = 1) by the EV-A71 strains C4 (green squares), C1-16 (pink triangles), and C1-06 (orange triangles) at 24 and 72 h pi. Echovirus 6 (Echo-6, purple) was included as a cytolytic control. MI, mock-infected BLECs. The data are expressed as the means of six independent replicates, with the error bars representing standard deviation. Statistical significance was determined with one-way ANOVA with multiple comparisons test. (B) Infected hBLECs and MI BLECs expressed as the percentage of cells showing immunofluorescence staining of replicative viral dsRNAs. For each virus strain, the data are expressed as the means ± standard deviation obtained from 10 confocal microscopy images analyzed with ImageJ software. Statistical significance was determined with the Kruskal–Wallis test. *$P \leq 0.05$; **$P \leq 0.01$; ****$P \leq 0.0001$.

lower compartments for the C1-16 (Fig. 2B) and C1-06 (Fig. 2C) viruses had roughly similar patterns to that of C4.

The release of infectious virus particles was examined in the upper compartment (Fig. 2D through F). For the C4 strain, extracellular infectious virus was detectable but not quantifiable at 6 hpi (Fig. 2D). The virus titer increased to a median of 6,500 MPNCU (most probable number of infectious units)/mL at 24 hpi and plateaued at 8,100 MPNCU/mL at 72 hpi. At 48 hpi, the median amount of the C1-16 virus (8,000 MPNCU/mL) released from the luminal side of infected hBLECs was similar to that of C4-infected hBLECs (Fig. 2E). Although the data had large standard deviations (Fig. 2F), hBLECs infected with

the C1-06 virus generated moderately high amounts of extracellular infectious virus (16,250 and 12,330 MPNCU/mL at 48 and 72 hpi, respectively). Virus titers determined for the Echo-6 lytic control at 24 hpi were around $10^6$ MPNCU/mL (Fig. S6B). The yield of the infectious virus was also determined in the lower compartment . Infectious virus was detected at all timepoints for the three EV-A71 strains but at levels below the quantitation limit of the titration assay. The overall data indicate that the EV-A71-infected hBLECs release moderate amounts of extracellular infectious virus and high amounts of extracellular viral RNAs from their luminal side, while moderate to low amounts of viral RNAs and non-quantifiable amounts of infectious virus are discharged within the lower compartment.

## Transendothelial migration of EV-A71-infected leukocytes

The experiments described above showed that cell-free EV-A71 particles display low capacity to replicate in hBLECs and to cross the BBB, so we investigated whether white blood cells could serve as transfer vehicles and allow EV-A71 to cross the BBB via transendothelial migration. First, we analyzed the susceptibility to virus strains of white blood cells isolated after whole red blood cell lysis ($n$ = 5 blood donors). The whole leukocyte fraction was inoculated (MOI = 1), and at 24 hpi, the leukocytes underwent fluorescence-activated cell sorting to separate monocytes (CD45+CD14+), neutrophils (CD45+CD15+), and NK/T (CD45+CD19−) and B (CD45+CD19+) cells. The RNA viral load

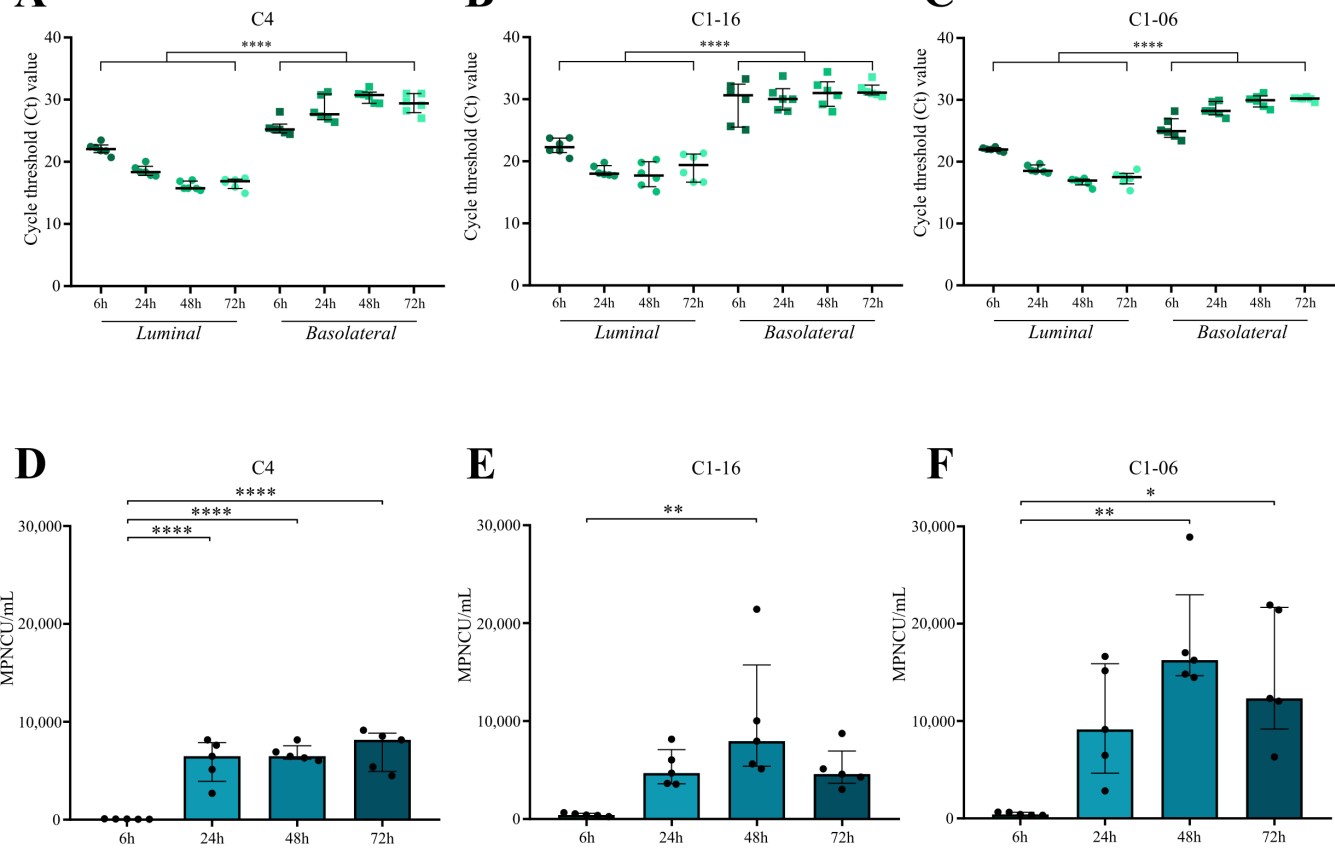

**FIG 2** Replication kinetics of three enterovirus A71 (EV-A71) strains in hBLECs. hBLECs were infected (MOI = 1) with the EV-A71 strains C4, C1-16, and C1-06 and analyzed at the indicated timepoints pi. (A–C) Positive viral RNA detected by semi-quantitative RT-qPCR in the luminal and basolateral compartments of hBLECs infected with EV-A71 strains C4 (A), C1-16 (B), and C1-06 (C). (D–F) Viral production determined in hBLECs infected with EV-A71 strains C1-06 (D), C4 (E), and C1-16 (F) by virus titration. The data are expressed as the most probable number of infectious units produced at 6, 24, 48, and 72 hpi. The data are the median with an interquartile range of six and five replicates, respectively, for RT-qPCR and titration. Statistical significance was determined with one-way ANOVA with the multiple comparisons test (A – E) and Kruskal–Wallis test (F). *$P \leq 0.05$; **$P \leq 0.01$; ****$P \leq 0.0001$.

was determined in each cell fraction by EV RT-qPCR (Fig. 3). Monocytes were associated with different EV-A71 RNA levels (median number of genome copies per cell, gc/c) in descending order C1-06 (225.5 gc/c), C1-16 (35 gc/c), and C4 (14.7 gc/c). The neutrophil fraction contained moderate amounts of EV-A71 RNA (from 67.5 gc/c to 28.3 gc/c). T and B cell fractions contained similar levels of C1-06 RNA (22.01 gc/c and 27.6 gc/c, respectively). In contrast, the amount of C1-16 RNA was six times higher in B cells (21.3 gc/c) than in T cells (3.5 gc/c), and the C4 RNA levels were equally low in lymphocytes. All leukocyte fractions contained minimal amounts of Echo-6 RNA (1.2 to 1.98 gc/c; Fig. 3D).

To perform the transendothelial migration experiment, the BBB ECs were grown on an insert with a 3.0-µm pore membrane, as previously described (27). For this investigation,

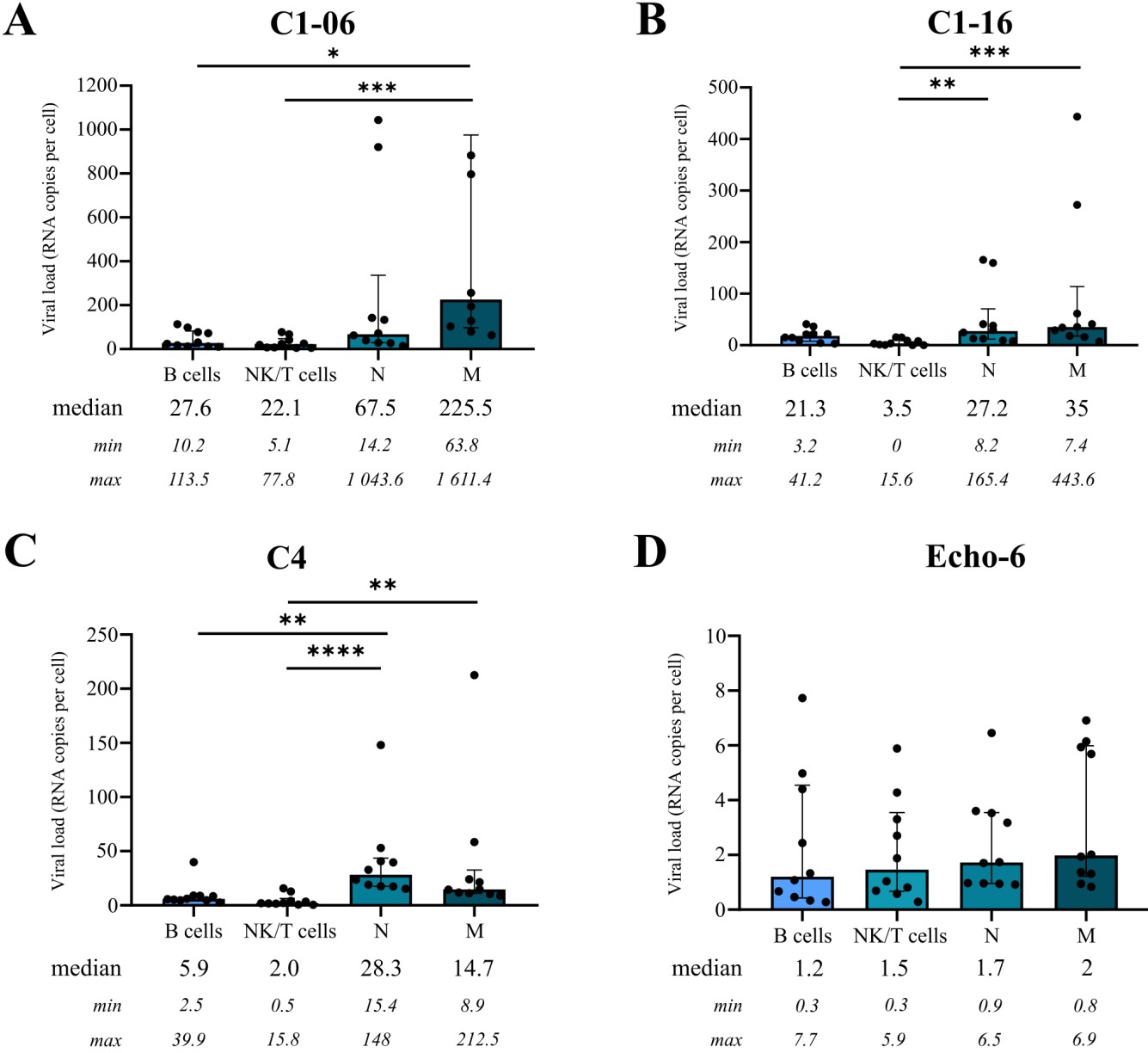

**FIG 3** Susceptibility of human blood leukocytes to enterovirus A71 and echovirus 6. Blood leukocytes were isolated from whole blood by the red blood cell lysis method ($n$ = 5 blood donors) and infected at MOI = 1 with the indicated virus strains. At 24 hpi, the leukocytes were separated by fluorescence-activated cell sorting to separate and count monocytes (CD45+CD14+), neutrophils (CD45+CD15+), and NK/T (CD45+CD19−) and B (CD45+CD19+) cells. Enterovirus RT-qPCR was performed on each leukocyte fraction to determine the viral load (viral RNA copies per cell) associated with the cell populations. Median, minimum (min), and maximum (max) are indicated in viral RNA copies per cell.

we restricted our comparison to C1-06 and C1-16 because we found limited differences between C1-16 and C4. The whole white blood cells ($n = 6$ blood donors) were isolated with the red blood cell lysis method, inoculated with EV-A71, and labeled with a fluorescent dye (CellTracker green). After 24 h of infection, they were placed on top of differentiated hBLECs to allow their adhesion and transmigration (Fig. 4A). After an incubation of 17 hours, hBLECs incubated with MI or infected leukocytes were imaged by confocal microscopy (Fig. 4B). For both conditions, attached leukocytes were analyzed on the upper side of hBLECs. Quantification of immune cells attached to the hBLEC surface showed that C1-16 induced significant adhesion compared with MI and C1-06 conditions (Fig. 4B). Measurements of the CellTracker fluorescence signal in the lower compartment indicated that MI and infected immune cells crossed hBLEC barriers (data not shown), and confocal microscopy showed immune cells embedded within hBLECs for both conditions (Fig. S7). Monocytes, B cells, NK/T cells, and neutrophils present in the lower transwell compartment were separated by fluorescence-activated cell sorting for analysis. After infection with EV-A71, the NK/T cells that transmigrated represented a greater percentage of the total cell content compared with MI immune cells (Fig. 4C). For immune cells infected with C1-16 and C1-06, the proportion of NK/T cells detected in the lower compartment increased to reach 25.6% (range, 15.8–48%) and 17.7% (range, 10.2–43.8%), respectively, compared with MI cells. The whole viral load (genome copy number, gc) was analyzed in each immune cell population harvested from the lower compartment (Fig. 4D). The whole gc levels were higher in cells infected with C1-16 than in those infected with C1-06. C1-16 was detected in all immune cell populations, except B cells. The C1-16 viral loads were similar in NK/T cells (median, 678 gc; range, 225–3631 gc) and neutrophils (median, 894 gc; range, 137–3524 gc), and it was highest in monocytes (median, 3060 gc; range, 631–4983 gc). In contrast, C1-06 was detectable in less than half of the samples analyzed, and the viral load was extremely low in neutrophils (median, 165 gc; range, 96–367 gc) and monocytes (median, 285 gc; range, 160–3476 gc). There was little evidence of infection and migration of B cells in the conditions tested. The whole viral load in the medium recovered in the lower compartment after centrifugation to exclude immune cells displayed a different pattern between the two EV-A71 strains (Fig. 4E). The C1-16 viral load (median, $1.7 \times 10^5$ gc; range, $1.3 \times 10^4$–$2.3 \times 10^6$ gc) was lower than that of C1-06 (median, $1.1 \times 10^6$ gc; range, $4.4 \times 10^4$–$1.6 \times 10^7$ gc) ($P < 0.05$), suggesting variations in the release of viral genomes from cells between C1-16 and C1-06. In the upper compartment, the amounts of viral RNAs were considerably higher than those detected in the lower compartment (Fig. S8), demonstrating again the role of the BBB in impeding the passage of viral RNA or virus particles. Together, the data indicate that different populations of EV-A71-infected immune cells cross the hBLEC barrier.

## DISCUSSION

There are three possible routes for viruses to cross the human BBB, the paracellular and transcellular pathways for cell-free virus particles and the migration of virus-infected immune cells (28). Using three phylogenetically distinct EV-A71 strains, including viruses of subgenogroups C1 and C4, and the emerging multirecombinant C1-like virus, we report that immune cells may serve as a transfer vehicle for EV-A71 to cross the human BBB.

First, we showed that our human BBB model provided a strong structural and functional roadblock to cell-free virus particles dispensed at the luminal side to mimic an infection from the bloodstream compartment. Our data agreed with those of our previous results using another human BBB model (hCMEC/D3) and those obtained with a 3D model of the BBB (primary human brain microvascular endothelial cells) (20, 29). Early after inoculation of hBLECs with EV-A71, we detected no substantial changes in TJ protein expression and localization nor a loss of TJ ultrastructural integrity, and at later timepoints, the paracellular tightness of the BBB was not compromised by EV-A71 replication within hBLECs. Accordingly, paracellular trafficking of virions is an unlikely

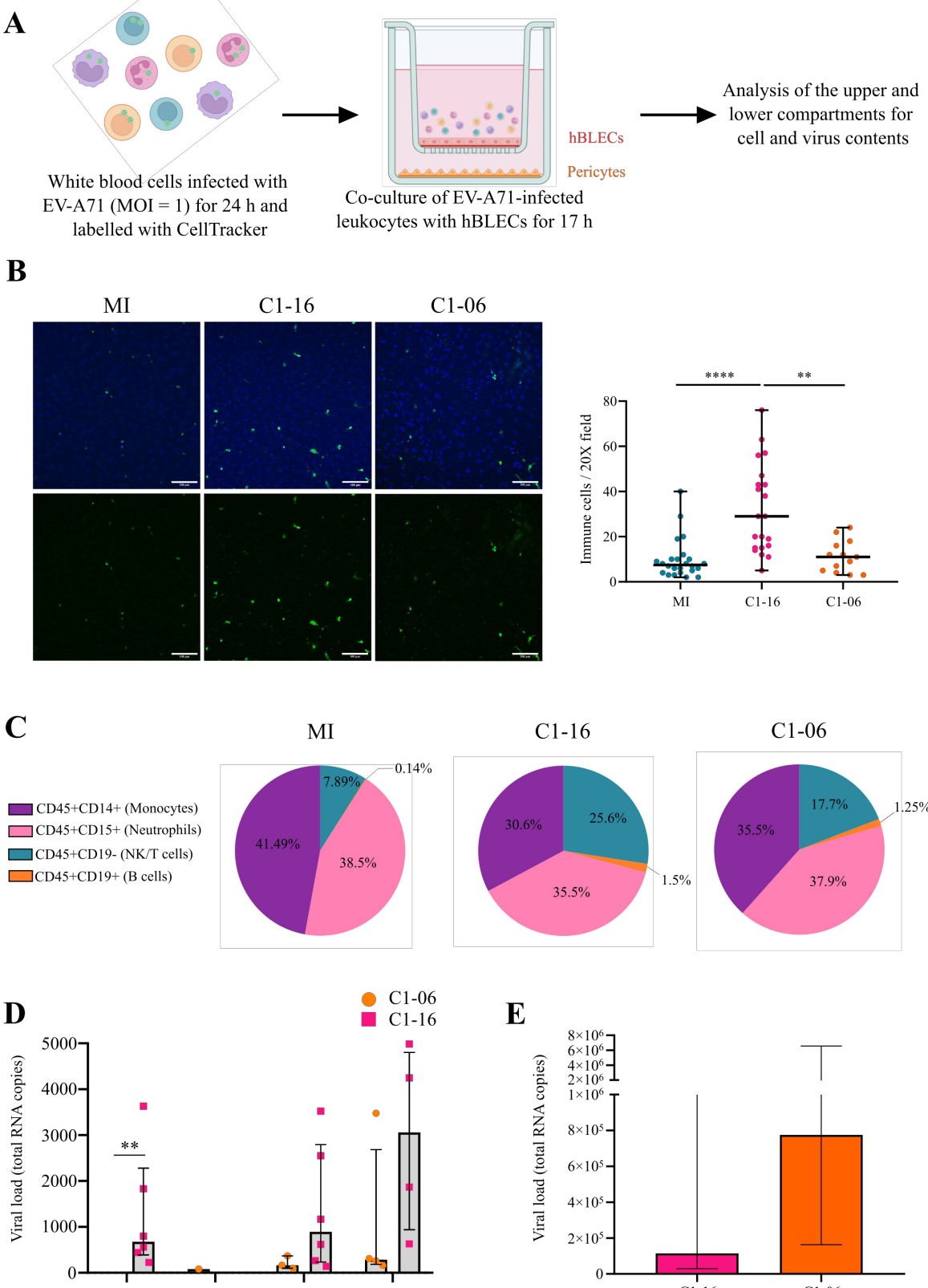

**FIG 4** Crossing of EV-A71-infected leukocytes through the human BBB. (A) Schematics of the transwell transmigration assay. Leukocytes (T and B cells, monocytes, and neutrophils) isolated from whole blood were mock-infected or infected with EV-A71 C1-06 or C1-16 (MOI 1) for 24 hours. For each condition, six independent experiments each corresponding to leukocytes from one blood donor were performed in triplicate. After extensive washing, leukocytes were

**FIG 4** (Continued)

labeled with CellTracker and added on top of a hBLEC barrier grown on a transwell membrane. Transmigration was allowed to proceed for 17 hours. The medium in the upper and lower transwell compartments was collected, hBLECs were fixed, and nuclei were labeled with DAPI. (B) Confocal microscopy analysis (merged images) was used to visualize the interaction of leukocytes (green) with hBLECs (nuclei in blue). Z-stack images were used to analyze the number of immune cells attached to the upper surface. Scale bar, 100 µm. The graphs for MI immune cells and C1-16 infected cells show the results of 24 and 21 independent image acquisitions, respectively from six blood donors. For C1-06-infected cells, the graph shows the results of 15 image acquisitions for five donors. Data are expressed as the median ±interquartile range. (C) After transmigration, leukocytes in the lower transwell compartment were harvested and prepared for fluorescence-activated cell sorting analysis (see legend to Fig. 3) to quantify the immune cells of the populations and conditions indicated. The pie charts indicate the proportion of each cell population. Data are expressed as the median ±interquartile range. Enterovirus RT-qPCR assay was performed in the leukocyte fraction (D) and the medium (E) recovered from the lower compartment to determine the total viral loads. Statistical significance was determined with the Kruskal–Wallis test (B and E) and a t test (D) **$P \leq 0.01$; ****$P \leq 0.0001$.

route for EV-A71 to pass through the BBB. The paracellular passage of pathogens occurs through disruption of TJs as a result of dysregulation of BBB integrity in response to the activity of inflammatory host immune factors or microbial factors on the microvascular ECs (30). A number of bacterial pathogens (*E. coli* K1, *S. pneumoniae*, *H. influenzae* type B, and *Streptococcus* spp) and viruses (HIV-1 mouse hepatitis virus) are considered to cross the BBB through paracellular transmigration (28, 31). This mechanism is not exclusive of transcellular migration, a second pathway that refers to the receptor-mediated penetration of viruses within ECs followed or not by active virus replication (24, 32, 33). Our data on EV-A71 indicate that the virus replicates moderately in hBLECs and, throughout the 3-day observation, the infected cells had focal distribution in one or a few adjacent cells. The low proportion of cells showing active viral replication early after inoculation and later, and the slow decline of cell viability are indicative of poor cell susceptibility to the virus. Three days after hBLEC inoculation, infectious progeny virus was released in low amounts from the luminal side of the BBB, indicating low permissivity of hBLECs. The released virions were unable to generate secondary infections because the proportion of infected hBLECs did not increase over time. Overall, our observation of poor susceptibility and permissivity of the human BBB after infection is consistent with data from immune-histochemistry and *in situ* hybridization investigations of human autopsy tissue samples showing few, if any, viral antigens and RNAs within the vascular endothelium (5). Our results on the strong resistance of hBLECs to EV-A71 contrast with data indicating that the barrier properties of human brain-like microvascular cells are disrupted by poliovirus and coxsackievirus B3 (34, 35). We found, however, that EV-A71 RNA and infectious viruses were released in the lower compartment as early as 6 hpi, at a time when virus production in the upper compartment is still low. This early passage through the hBLECs did not vary over time. This finding indicates that almost all the EV-A71 progeny was released unidirectionally from the luminal side of hBLECs, and therefore the passage of cell-free EV-A71 across the human BBB is limited. In the literature, evidence suggests that viral RNA and virus particles of EV-A71 can be produced and transmitted within small extracellular vesicles in rhabdomyosarcoma cells and that EV-A71 trapped in small extracellular vesicles can cross an *in vitro* model of the human BBB (36, 37). Accordingly, we cannot exclude that the EV-A71 stocks used in our study contained small extracellular vesicles that delivered viral RNAs and virus particles to the basolateral side of hBLECs.

Another reported pathway for viruses to reach the human brain parenchyma is via paracellular (or even transcellular) diapedesis of infected immune cells trafficking from the blood across the BBB (38, 39). Several RNA viruses bypass the human BBB to enter the CNS through transendothelial transport via diapedesis of infected immune cells (40, 41). Previous studies reported that EV-A71 replicates in human monocytes and lymphocytes and in rhesus monkey CD14[+] cells (42, 43). In our study, we provided evidence of the passage through hBLECs of white blood cells infected by EV-A71 and determined that attachment of immune cells to hBLECs infected with the C1-like virus (C1-16) was higher than that of immune cells infected with C1-06. Findings showing increased adhesion to endothelial cells of infected T cells and monocytes have been reported for two

flaviviruses, Usutu and West Nile viruses (39, 44). Our detection of viral RNA in infected immune cells that transmigrated across hBLECs indicated that neutrophils, monocytes, and NK/T cells were infected. Monocytes, neutrophils, and NK/T cells infected with EV-A71 were not impaired in their transmigration properties through hBLECs. Our data suggested that EV-A71 infection enhanced the migration of NK/T cells, while migration of monocytes and neutrophils remained unchanged. However, we observed marked variations among replicates. Although C1-16 adhered better to hBLECs than C1-06 and the rates of T cell transmigration might be higher with C1-16 than C1-06, the differences were not sufficient to counterbalance the active replication of C1-06 in leukocytes, thereby explaining the different RNA levels detected in the basolateral compartment of the BBB model at the end of the transmigration assay. Our data are in line with recent findings suggesting the involvement of immune T cells in the pathogenesis of pediatric EV-A71 infections (45).

Earlier studies indicated that human dendritic cells transmit EV-A71 and could be involved in EV-A71 infectious disease (46, 47). In the present study, we showed that immune cells such as monocytes, neutrophils, and NK/T cells can be highjacked by EV-A71 and transmit the virus across a human BBB barrier restrictive to cell-free EV-A71 virions, a mechanism which can also potentially contribute to EV-A71 pathogenesis.

## MATERIALS AND METHODS

### Virus strains used in the study

Three clinical isolates of EV-A71 designated C1-06, C1-16, and C4 and an echovirus 6 isolate (Echo-6) were used in this study (Table S1). The virus strains were propagated in RD cells, and the virus stocks were checked as described in the Supplemental Material.

### Co-cultures of ECs derived from CD34$^+$ hematopoietic stem cells and brain pericytes

The ECs are derived from CD34$^+$ hematopoietic progenitor cells isolated from umbilical cord blood (48). Cultures of ECs and brain pericytes were expanded in 100-mm Petri dishes coated previously with 0.2% porcine gelatin (Merck). The ECs were cultured as previously reported (22). The CD34$^+$-derived ECs were cultured in the EC medium (ECM, ScienCell) containing 50 µg/mL gentamicin (Sigma-Andrich) and supplemented with EC growth supplement 1% (ECGS; ScienCell) and heat-inactivated fetal bovine serum 5% (FBS; Sigma-Aldrich). The medium is designated ECM-5. The pericytes were cultured in high glucose Dulbecco's modified Eagle medium (DMEM; Dutscher) supplemented with 20% FBS, 1% glutamine, and 1% streptomycin–penicillin (DMEM-20).

### Preparation of *in vitro* BBB in Transwell inserts

The human CD34$^+$-derived ECs were differentiated into human brain-like ECs (hBLECs) with pericytes to generate an *in vitro* BBB model (22, 23). To obtain the BBB, the CD34$^+$-derived ECs were cultured upward of the semi-permeable membrane in a transwell insert (12-well plates, 0.4 µm, Costar #3460), and pericytes were grown downward. The transwell inserts were flipped over and coated with rat collagen I (10 µg/cm$^2$) for 1 hour at 37°C before the addition of brain pericytes ($4.46 \times 10^4$ cells/cm$^2$). After 3 hours at 37°C, the inserts were flipped over in 12-well plates containing an endothelial cell growth medium MV2 (ECGMV2, Promocell) (1.5 mL per well). The upper surface of the membrane was precoated with Matrigel (Corning, #356237), and after 1 hour at room temperature, CD34$^+$-derived ECs in ECGMV2 were seeded at $7.14 \times 10^4$ cells/cm$^2$. The culture medium was changed every 2 days during 6 days of co-culture, after which ECs were fully differentiated into hBLECs and could be used for further experiments (22, 49).

The *in vitro* BBB model was adapted for transmigration tests, as described previously (27). Briefly, CD34$^+$-ECs were cultured in 3.0 µm pore membrane inserts (Costar #3462),

as described above, except that pericytes and the medium were absent in the lower compartment. One week later, inserts were transferred in 12-well plates containing brain pericytes to initiate differentiation into hBLECs. Transmigration experiments were performed 5 days later.

## Assessment of BBB permeability

BBB integrity was assessed by testing the paracellular permeability to LY (Sigma Aldrich), a 400-Da hydrophilic fluorescent compound. The transwell inserts with fully differentiated ECs (i.e., hBLECs) were transferred to new 12-well plates containing warm (37°C) Hank's Balanced Salt Solution (HBSS, Sigma Aldrich) supplemented with 1% sodium pyruvate (Sigma Aldrich) and 1% HEPES buffer 1M (Dutscher). The medium in the upper compartment was replaced by 500 µL of a warm 50 µM LY solution (in HBSS). The transwell inserts were transferred every 20 minutes to a new well containing warm HBSS solution for 1 hour. The intensity of LY fluorescence in the lower compartments at each timepoint and in the upper compartments was determined with a Fluoroscan analyzer (Thermo Scientific) at excitation/emission wavelengths of 432/538 nm. The clearance principle was applied to calculate the permeability coefficient (Pe, in $cm.min^{-1}$) of the BBB, a concentration-independent transport parameter. Pe was calculated as reported elsewhere (50). Briefly, the amount of LY in the lower compartment was divided by the concentration of the initial solution to determine the cleared volume, which was then plotted versus time to calculate the slope by linear regression analysis. The filter surface permeability (PSf) was calculated from the slope of the clearance curve obtained from control devices, i.e., membrane with coatings plus brain pericytes. The total surface permeability (PSt) was calculated from the slope of the clearance curve determined from co-culture devices. The EC surface Pe coefficient (PSe) was calculated as follows [1/PSe = 1/PSt − 1/PSf] divided by the membrane surface area (1.12 $cm^2$ for 12-well transwell inserts).

## Transmigration of infected white blood cells across the BBB

White blood cells were isolated from blood samples obtained from blood donors at the Etablissement Français du Sang (EFS), a national public transfusion service. Fresh blood was incubated with 1X lysis buffer (BD Pharm Lyse) for 10–15 minutes, and lysis was stopped by adding PBS (vol/vol). After centrifugation (300 g, 10 min), the pellet was washed twice with PBS, and leukocytes were numbered with an XN-2000 hematology analyzer (Sysmex). Leukocytes ($5 \times 10^5$ cells) were diluted in RPMI medium and inoculated (MOI 1) for 1 hour and 30 minutes at 37°C. Cells were labeled with CellTracker Green (Invitrogen) according to the manufacturer's instructions. The cell suspension was centrifuged (300 g, 8 min) to remove unbound virions, and cells were incubated at 37°C in RPMI with 1% heat-inactivated bovine fetal serum (BFS) for 24 hours in a $CO_2$ incubator. After 24 hours, infected leukocytes were transferred on top of hBLECs (prepared as described above) for a transmigration assay. The medium in the bottom chamber was replaced with RPMI with 5% heat-inactivated BFS and 100 ng/mL of MCP-1 (Miltenyi Biotech). White blood cells were co-cultured with hBLECs for 17 hours and recovered from the top and bottom chambers of the insert. White blood cells were analyzed by fluorescence-activated cell sorting. To detect cell surface markers, leukocyte cells were marked with a cocktail of primary-labeled antibodies (Table S2) for 20 minutes at room temperature. Cells were analyzed with a 5-channel flow cytometer BD FACSAria SORP (Becton Dickinson) to separate neutrophils ($CD45^+CD15^+$), monocytes ($CD45^+CD14^+$), B ($CD45^+CD19^+$) cells, and NK/T ($CD45^+CD19^-$) cells. Separated cells were resuspended in 200 µL of lysis EasyMag buffer for nucleic acid extraction and viral RNA detection. CellTracker-labeled leukocytes were imaged with hBLECs by confocal fluorescence microscopy.

## Confocal fluorescence microscopy and imaging

hBLECs were washed twice with phosphate-buffered saline (PBS) and conserved at 4°C in PBS for a maximum of 1 week. The cell monolayers were fixed by two methods. To detect ZO-1 protein, fixation was performed for 15 minutes at room temperature with a 4% paraformaldehyde solution followed by permeabilization with 0.1% Triton X-100 for 10 minutes. To detect CLDN-5 protein, fixation was done with ice-cold methanol for 30 seconds. Three target markers were analyzed with the following primary antibodies: ZO-1 (Cell Signaling D6L1E #13663; dilution 1:400), CLDN-5 (Invitrogen #34–1600; 1:400), and viral double-stranded (ds)RNA (Scicons #10010200; 1:1000). The primary antibodies were diluted in 4% BSA prepared in PBS and incubated overnight at 4°C. After washing three times (0.2% Tween 20, 1% BSA in PBS), the cells were incubated for 1 hour at room temperature with a fluorochrome-coupled secondary antibody (dilution 1:500 in 4% BSA solution): nuclei were counterstained with DAPI (1:1000). The membranes were carefully cut from transwell inserts, mounted on glass slides, and covered with VECTASHIELD Antifade Mounting Medium (Invitrogen). Images were acquired with a LSM800 Leica microscope and processed with ImageJ software.

## Enterovirus genome quantification by RT-PCR

Semi-quantitative assay. The total nucleic acids were extracted from 200 µL of infected hBLEC samples with the Maxwell RSC Blood Kit and Maxwell RSC instrument (Promega). The quantification of viral genomes in the samples was performed by the commercial semi-quantitative real-time RT-PCR assay "Enterovirus R-GENE kit" (bioMerieux, Marcy l'Etoile, France) and the CFX Opus 96 Real-Time PCR System (Bio-Rad).

## Quantitative assay for samples of infected white blood cells

Total nucleic acids were extracted in 96-well microplates with the automated workstation MGISP-960 (MGI Tech) and MGIEasy Nucleic Acid Extraction kit. Total viral RNA was quantified in infected white blood cells with a lab-developped-RT-qPCR assay that amplifies a 148-base pair genome target (51). Amplification was performed with the Master Mix RT-qPCR TaqPath1-step CG kit (Thermofisher) and the QuantStudio thermocycler.

## Statistical analysis

Statistical analysis was conducted using GraphPad Prism software version 7.04 (GraphPad Software, San Diego,CA, USA). All data are shown as mean or median ±SEM. Kruskal–Wallis test, one-way ANOVA with multiple comparisons, and unpaired Student's $t$-tests were used to determine significance, as appropriate. When data had a normal distribution (D'Agostino–Pearson test), parametric tests were applied. Otherwise, nonparametric tests were used. $P$ values $< 0.05$ were considered significant and are indicated in figure legends.

### ACKNOWLEDGMENTS

We thank Isabelle Simon (LMGE), Arnaud Lapeyre (LMGE), and Gwendoline Jugie (LMGE) for technical assistance in virus titration and sequencing. The authors gratefully acknowledge Caroline Vachias from the Clermont Imagerie Confocale (CLIC) platform and Mélanie Soucal and Clara Tournebize from the platform of flow cytometry and cell sorting (CHU Clermont-Ferrand) for their support and assistance in this work and Pierre Sauvanet (M2iSH) for his helpful advice.

This work was funded in part by OrganoVIR (Grant 812673) of the European Union's Horizon 2020 programme.

Léa Gaume, Data curation, Investigation, Visualization, Methodology, Writing–original draft, Writing–review and editing | Hélène Chabrolles, Investigation, Methodology | Maxime Bisseux, Data curation, Investigation | Igor Lopez-Coqueiro, Investigation,

Methodology | Lucie Dehouck, Resources, Methodology | Audrey Mirand, Resources, Writing–review and editing | Cécile Henquell, Methodology, Funding acquisition, Writing–review and editing | Fabien Gosselet, Resources, Data curation, Methodology, Writing–review and editing | Christine Archimbaud, Data curation, Writing–review and editing, Conceptualization | Jean-Luc Bailly, Writing–review and editing, Conceptualization, Funding acquisition, Project administration.

## AUTHOR AFFILIATIONS

[1]Laboratoire Microorganismes: Génome et Environnement (LMGE), CNRS UMR 6023, Clermont Auvergne Université, Clermont-Ferrand, France

[2]Laboratoire de Virologie, Centre National de Référence des Entérovirus et Parechovirus, CHU Clermont-Ferrand, Clermont-Ferrand, France

[3]Laboratoire de la Barrière Hémato-Encéphalique (LBHE), Université d'Artois, Lens, France

## AUTHOR ORCIDs

Léa Gaume http://orcid.org/0009-0007-9406-7538
Hélène Chabrolles http://orcid.org/0000-0001-8038-2130
Maxime Bisseux http://orcid.org/0000-0002-3489-049X
Cécile Henquell http://orcid.org/0000-0003-2859-6566
Fabien Gosselet http://orcid.org/0000-0002-0481-5026
Christine Archimbaud http://orcid.org/0000-0001-7665-5618
Jean-Luc Bailly http://orcid.org/0000-0002-9770-6450

## FUNDING

| Funder | Grant(s) | Author(s) |
|---|---|---|
| EC | Horizon 2020 Framework Programme (H2020) | OrganoVIR Grant 812673 | Jean-Luc Bailly |

## DATA AVAILABILITY

All data generated or analyzed during this study are included in the article.

## ETHICS APPROVAL

Written and informed consent from the donor's parents was obtained for the collection of umbilical cord blood, in compliance with French legislation (CODECOH number, DC-2011-1321). The ECs are derived from CD34+ hematopoietic progenitor cells isolated from umbilical cord blood (48). PBMCs were obtained from healthy adult donors at the Etablissement Français du Sang (EFS) with the agreement reference EFS AURA 22-017. In France, collection, qualification, preparation, and distribution of labile blood products for civilians are under the exclusive responsibility of the national public transfusion service EFS. Written informed consent for research use was obtained at the EFS.

## ADDITIONAL FILES

The following material is available online.

### Supplemental Material

**Supplemental material (Spectrum00690-24-s0001.pdf).** Fig. S1-S8; Tables S1 and S2.

### Open Peer Review

**PEER REVIEW HISTORY (review-history.pdf).** An accounting of the reviewer comments and feedback.

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
