## [Reviewer comments · Microbiology Spectrum]

Microbiology Spectrum

Enterovirus A71 crosses a human blood-brain barrier model through infected immune cells

Léa Gaume, H el ene Chabrolles, maxime Bisseux, Igor Coqueiro, Lucie Dehouck, Audrey Mirand, C ecile Henquell, Fabien Gosselet, Christine Archimbaud, and Jean-Luc Bailly

Corresponding Author(s): Jean-Luc Bailly, Universit e Clermont Auvergne

Review Timeline:

Submission Date:	March 16, 2024
Editorial Decision:	April 3, 2024
Revision Received:	April 16, 2024
Accepted:	April 16, 2024

Editor: Peter Pelka

Reviewer(s): Disclosure of reviewer identity is with reference to reviewer comments included in decision letter(s). The following individuals involved in review of your submission have agreed to reveal their identity: Hiroyuki Shimizu (Reviewer #3)

Transaction Report:

DOI: <https://doi.org/10.1128/spectrum.00690-24>

Re: Spectrum00690-24 (Enterovirus A71-infected immune cells cross a human brain-like endothelial cell barrier)

Dear Dr. Jean-Luc Bailly:

Thank you for the privilege of reviewing your work. Below you will find my comments, instructions from the Spectrum editorial office, and the reviewer comments.

I have now received reviewer comments, which are appended below. The reviewers raised several concerns that need to be addressed before the manuscript can be published. I encourage you to make the appropriate revisions, with the requested additional experiments, prior to resubmission.

Revision Guidelines

Sincerely,
Peter Pelka
Editor
Microbiology Spectrum

Reviewer #1 (Comments for the Author):

The manuscript proposed by Gaume et al: "Enterovirus A71-infected immune cells can cross a human blood-brain barrier model" is concise, nicely written, and easy to understand. The manuscript addresses the enterovirus A71 ability to cross an in

vitro model of brain endothelium for reaching the central nervous system. Authors show that three clinical variants of EV-A71 poorly infect their brain endothelium model and do not severely damage it. To support the crossing of their in vitro model, authors explored the possibility that EV-A71 may infect leucocytes and use them as a Trojan horse.

Major comments

- (1) Authors indicate that the in vitro model of blood brain barrier is constituted of a layer of human brain-like endothelial cells (on transwell membrane) and brain pericytes (on well bottom). However the susceptibility of pericytes to EV-A71 (and Echovirus 6 control) is not presented.
- (2) The role of the Echovirus 6 as positive control for infection and lysis of endothelial cells is not enough presented in the text and its formulation is misleading (E6-01). As EV-A71 strains are named C1-06, C1-16, and C4; E6-01 may be confused with a 4th EV-A71 strain.

Optional comments

- Due to the difficulty and time necessary for isolating well-characterized leucocyte populations, the following comments are proposed as improvement of the manuscript and not as a sine qua non condition.
- (3) Authors infect different leukocytes populations by EV-A71, but are the leucocytes productively infected and can release viral progenies (not shown)?
 - (4) Can the Echovirus 6 control infects leucocytes as well? If not, it would highlight two clearly different crossing phenotypes among enteroviruses, direct infection vs Trojan horse.

Minor comments

- When talking about copies/ml, authors should precise RNA copies/ml
- Introduction line 83, authors indicate that immortalized cell line are associated with a number of limitations. Which ones?
- Results line 95, authors indicate that undifferentiated CD34+ EC cells are moderately susceptible. How much is "moderately"?
- MI abbreviation should be defined (mock infected), example line 156.

Reviewer #2 (Comments for the Author):

This paper describes the lack of migration of enterovirus A71 crossing the blood-brain model. Instead, the immune cells (monocytes, neutrophils, and T cells) were infected by EV-A71 and transmit it across human endothelial cells

Major comments

1. The immune cells still cross the BBB model whether it is infected with the virus. Therefore, the title is misleading.
2. The genome copies should be normalized with the number of immune cells since different amounts of immune cells crossed the BBB.
3. The results differences between C1-06 and C1-16 were not stated. Why both results are so different? Did C4 show similar results as C1-06? Should these results be presented together as EV-A71 instead of different strains?
4. Was Echo 6 used for infection of immune cells? Results should be presented.

Minor comments

1. The method section should be before the results? Some abbreviations appeared later in methods.
2. State the MOI used for the virus in the method. Provide accession number of the viruses.
3. For easier reading, it will be helpful to explain the LY, tight junction markers, permeability etc
4. Line 29-30. EV-A71 was first discovered in USA, and many outbreaks occurred in Asia and Europe.
5. Line 47- change youngest to young.
6. Correct the reference in line 70.

Reviewer #3 (Comments for the Author):

In this paper, Gaume and colleagues aimed to apply a cell culture-based human BBB model using human brain-like endothelial cells (BLECs) to investigate the neuroinvasion mechanism of enterovirus A71 (EV-A71) strains to CNS tissues. The topics addressed in this study would be critical and worthwhile. However, there are several methodological and/or presentation flaws in this paper, especially for immune cell involvement through BBB (Fig.3), as follows.

Specific comments

1. Virus replication kinetics of the three EV-A71 strains in RD cells should be described or cited.
2. Virus replication kinetics of EV-A71 strains in leukocytes without BLECs should be described (Fig. 3).
3. Fig. 3B; As far as I can see in Fig. 3B of the provided PDF file, the number of EV-A71-C1-16-infected leukocyte cells

(CellTracker-positive cells; green) attached with BLECs seem not to be much different with those for MI and C1-06. But please ignore any misunderstanding on my interpretation.

4. Fig. 3D; For the comparison, viral RNA from the upper compartment should be also assessed.

5. Page 7, lines 185-188; for immune cell transmigration assay, only two EV-A71 strains (C1 and C1-like) were used instead of three strains (Fig.3).

The manuscript proposed by Gaume et al: "Enterovirus A71-infected immune cells can cross a human blood-brain barrier model" is concise, nicely written and easy to understand. The manuscript addresses the enterovirus A71 ability to cross an in vitro model of brain endothelium for reaching the central nervous system. Authors show that three clinical variants of EV-A71 poorly infect their brain endothelium model and do not severely damage it. To support the crossing of their in vitro model, authors explored the possibility that EV-A71 may infect leucocytes and use them as a Trojan horse.

Major comments

(1) Authors indicate that the in vitro model of blood brain barrier is constituted of a layer of human brain-like endothelial cells (on transwell membrane) and brain pericytes (on well bottom). However the susceptibility of pericytes to EV-A71 (and Echovirus 6 control) is not presented.

(2) The role of the Echovirus 6 as positive control for infection and lysis of endothelial cells is not enough presented in the text and its formulation is misleading (E6-01). As EV-A71 strains are named C1-06, C1-16, and C4; E6-01 may be confused with a 4th EV-A71 strain.

Optional comments

Due to the difficulty and time necessary for isolating well-characterized leucocyte populations, the following comments are proposed as improvement of the manuscript and not as a sine qua non condition.

(3) Authors infect different leukocytes populations by EV-A71, but are the leucocytes productively infected and can release viral progenies (not shown)?

(4) Can the Echovirus 6 control infects leucocytes as well? If not, it would highlight two clearly different crossing phenotypes among enteroviruses, direct infection vs Trojan horse.

Minor comments

- When talking about copies/ml, authors should precise RNA copies/ml

- Introduction line 83, authors indicate that immortalized cell line are associated with a number of limitations. Which ones?

- Results line 95, authors indicate that undifferentiated CD34+ EC cells are moderately susceptible. How much is "moderately"?

- MI abbreviation should be defined (mock infected), example line 156.

Replies to the reviewers' comments

Reviewer #1 (Comments for the Author):

The manuscript proposed by Gaume et al: "Enterovirus A71-infected immune cells can cross a human blood-brain barrier model" is concise, nicely written, and easy to understand. The manuscript addresses the enterovirus A71 ability to cross an in vitro model of brain endothelium for reaching the central nervous system. Authors show that three clinical variants of EV-A71 poorly infect their brain endothelium model and do not severely damage it. To support the crossing of their in vitro model, authors explored the possibility that EV-A71 may infect leucocytes and use them as a Trojan horse.

Major comments

(1) Authors indicate that the in vitro model of blood brain barrier is constituted of a layer of human brain-like endothelial cells (on transwell membrane) and brain pericytes (on well bottom). However, the susceptibility of pericytes to EV-A71 (and Echovirus 6 control) is not presented.

Reply. Correct. We tested the susceptibility of the brain pericytes used in the model and found no viral replication in these cells. We have indicated this in the revised version of the manuscript (lines 100 – 101), as follows: “The susceptibility to EV-A71 of brain pericytes was also tested: these cells did not support EV-A71 RNA replication (data not shown).”

(2) The role of the Echovirus 6 as positive control for infection and lysis of endothelial cells is not enough presented in the text and its formulation is misleading (E6-01). As EV-A71 strains are named C1-06, C1-16, and C4; E6-01 may be confused with a 4th EV-A71 strain.

Reply. We thank the reviewer for this comment. We have made changes (lines 114 – 116) in the revised manuscript to describe the role of echovirus 6 as positive control: “In addition to EV-A71 strains, we tested an echovirus 6 strain, hereafter designated Echo-6. We included this virus as a cytolytic control of ECs known to replicate actively in hCMEC/D3 cells and disrupt an in vitro BBB (20).” We have renamed the echovirus 6 strain abbreviating it to “Echo-6” to distinguish clearly the virus designations and made the changes throughout the revised manuscript.

Optional comments

Due to the difficulty and time necessary for isolating well-characterized leucocyte populations, the following comments are proposed as improvement of the manuscript and not as a sine qua non condition.

(3) Authors infect different leukocytes populations by EV-A71, but are the leucocytes productively infected and can release viral progenies (not shown)?

Reply. We thank the reviewer for this comment. We chose to focus on the analysis of the replication of viral genomes in leucocytes. In an ongoing study, we are exploring the critical point (among others) of the production of infectious viruses in leucocyte populations.

(4) Can the Echovirus 6 control infect leucocytes as well? If not, it would highlight two clearly different crossing phenotypes among enteroviruses, direct infection vs Trojan horse.

Reply. Yes, we have included echovirus 6 in our experiments investigating the susceptibility of leucocytes to virus strains. Our results indicated that the production of echovirus 6 RNA was in the range of 0.03 to 1.98 genome copies per cell. We have interpreted this result as an indication of low, if any, virus replication in leucocytes and of a major difference between echovirus 6 and EV-A71. These data are now included in the new figure 3 and described in lines 169 and 170 as follows: “*All leucocyte fractions contained minimal amounts of Echo-6 RNA (1.2 to 1.98 gc/c; Fig. 3D).*”

Minor comments

- When talking about copies/ml, authors should precise RNA copies/ml

Reply. Done (changes made in lines 71 to 73).

- Introduction line 83, authors indicate that immortalized cell line are associated with a number of limitations. Which ones?

Reply. The immortalized cell line lost a number of endothelial characteristics (gaps in the expression of certain transporters and metabolic enzymes compared with in vivo conditions). We have developed this point in the revised manuscript in lines 84 to 86.

- Results line 95, authors indicate that undifferentiated CD34+ EC cells are moderately susceptible. How much is "moderately"?

Reply. At the request of reviewer 3 (comment 1), we have described the proportion of RD cells and CD34+ ECs infected with the EV-A71 strains at 24 hpi and provided the data in the new figure S2. We found that the proportions of infected CD34+ ECs were 5- to 13.2-fold lower than those of infected RD cells. We have mentioned the results in lines 98 – 100 of the revised manuscript: “*Compared with the highly susceptible rhabdomyosarcoma (RD) cells, the proportions of CD34+-ECs infected with EV-A71 were 5- to 13.2-fold lower than those of infected RD cells (Fig. S2).*”

- MI abbreviation should be defined (mock infected), example line 156.

Reply. Done. We have indicated the abbreviation at the first mention of “mock-infected” (line 102 of the revised manuscript).

Reviewer #2 (Comments for the Author):

This paper describes the lack of migration of enterovirus A71 crossing the blood-brain model. Instead, the immune cells (monocytes, neutrophils, and T cells) were infected by EV-A71 and transmit it across human endothelial cells.

Major comments

1. The immune cells still cross the BBB model whether it is infected with the virus. Therefore, the title is misleading.

Reply. Thank you for the comment. We have modified the title of the manuscript as follows: “*Enterovirus A71 crosses a human blood-brain barrier model through infected immune cells*”

2. The genome copies should be normalized with the number of immune cells since different amounts of immune cells crossed the BBB.

Reply. The number of cells crossing the BBB is very low compared with the number of cells dispensed into the upper compartment, and varied widely between the blood donors. These experimental variations were behind our choice to present the data in terms of overall viral load instead of viral load per cell; we thought that normalization with the number of cells would be misleading in this case.

3. The results differences between C1-06 and C1-16 were not stated. Why both results are so different? Did C4 show similar results as C1-06? Should these results be presented together as EV-A71 instead of different strains?

Reply. Thank you for this comment. We agree with the reviewer that we did not explain the differences in our results between the C1-06 and C1-16 strains. We hope this is now rectified by the inclusion of additional data in the revised manuscript (reported in new Fig. 3 and new Fig. S8). The data from Fig. 3 (*i.e.*, virus replication in white blood cells without hBLECs) showed clearly that C1-06 replicates better than C1-16. Among the four immune cell populations analyzed in our study, this difference is clearly seen in monocytes and to some extent in neutrophils and T cells. With the new Fig. S8 (*i.e.*, viral loads determined in the medium and the cell fraction recovered from the upper compartment at the end of the transmigration assay), we draw the same conclusion regarding the differences between the two EV-A71 strains. In the medium, the C1-06-infected leucocytes released as much as 10 times more viral RNA than the C1-16-infected leucocytes. This difference was also seen in the bulk of immune cells in the upper compartment. Combining the data presented in the revised manuscript, we came to the conclusion that the two EV-A71 strains differed in their replication in white blood cells as a whole: the levels of viral RNA generated in and released from monocytes, neutrophils and T cells infected with C1-06 are higher than with C1-16. Although C1-16 adhered better to hBLECs than C1-06 (Fig. 4B) and the rates of T cell transmigration might be higher with C1-16 than C1-06 (Fig. 4C), these differences were not sufficient to counterbalance the active replication of C1-06 in leucocytes, thereby explaining the different RNA levels detected in the basolateral compartment of the BBB model at the end of the transmigration assay. Investigations are ongoing to address the underlying factors contributing to these variations. In the revised manuscript, we have explained the differences between virus strains in lines 262 – 266.

4. Was Echo 6 used for infection of immune cells? Results should be presented.

Reply. We thank the reviewer for this comment. We have included the results of our experiments investigating the susceptibility of leucocytes in the new figure 3. See also the reply to comment 4 of reviewer 1.

Minor comments

1. The method section should be before the results? Some abbreviations appeared later in methods.

Reply. We found no constraint on this point in the recommendations to authors. Accordingly, we have left the section order unchanged and checked the abbreviations throughout.

2. State the MOI used for the virus in the method. Provide accession number of the viruses.

Reply. Done. We have checked the manuscript and found that MOIs were indicated in the section “materials and methods” and the legends to the figures. Accession numbers indicated in Table S1.

3. For easier reading, it will be helpful to explain the LY, tight junction markers, permeability etc

Reply. Done. We have added technical details in lines 104 – 106 for easier understanding.

4. Line 29-30. EV-A71 was first discovered in USA, and many outbreaks occurred in Asia and Europe.

Reply. Done.

5. Line 47- change youngest to young.

Reply. Done.

6. Correct the reference in line 70.

Reply. We have mentioned reference 16 at the end of the sentence.

Reviewer #3 (Comments for the Author):

In this paper, Gaume and colleagues aimed to apply a cell culture-based human BBB model using human brain-like endothelial cells (BLECs) to investigate the neuroinvasion mechanism of enterovirus A71 (EV-A71) strains to CNS tissues. The topics addressed in this study would be critical and worthwhile. However, there are several methodological and/or presentation flaws in this paper, especially for immune cell involvement through BBB (Fig.3), as follows.

Specific comments

1. Virus replication kinetics of the three EV-A71 strains in RD cells should be described or cited.

Reply. We have reported the data showing EV-A71 replication in RD cells in the new figure S2 of the supplemental material. We have described the results in lines 98 – 100 of the revised manuscript as follows: “*Compared with the highly susceptible rhabdomyosarcoma (RD) cells, the proportions of CD34+-ECs infected with EV-A71 were 5- to 13.2-fold lower than those of infected RD cells (Fig. S2).*”

2. Virus replication kinetics of EV-A71 strains in leukocytes without BLECs should be described (Fig. 3).

Reply. We have determined the levels of virus replication in immune cells at 24 hpi without BLECs and reported the data in the new figure 3. We have described the results in lines 158 – 170 of the revised manuscript: “*First, we analyzed the susceptibility to virus strains of white blood cells isolated after whole red blood cell lysis (n = 5 blood donors). The whole leukocyte fraction was inoculated (MOI = 1) and at 24 hpi, the leukocytes underwent fluorescence-activated cell sorting to separate monocytes (CD45+CD14+), neutrophils (CD45+CD15+), and NK/T (CD45+CD19+) and B (CD45+CD19+) cells. The RNA viral load (genome copy number per cell, gc/c) was determined in each cell fraction by EV RT-qPCR (Fig. 3). Monocytes*

were associated with different EV-A71 RNA levels (median number of genome copies per cell, gc/c), in descending order C1-06 (225.5 gc/c), C1-16 (35 gc/c), and C4 (14.7 gc/c). The neutrophil fraction contained moderate amounts of EV-A71 RNA (from 67.5 gc/c to 28.3 gc/c). T and B cell fractions contained similar levels of C1-06 RNA (22.01 gc/c and 27.6 gc/c, respectively). In contrast, the amount of C1-16 RNA was 6 times higher in B cells (21.3 gc/c) than in T cells (3.5 gc/c) and the C4 RNA levels were low in lymphocytes. Minimal amounts of Echo-6 RNA (1.2 to 1.98 gc/c) were found in leucocytes (Fig. 3D)."

3. Fig. 3B; As far as I can see in Fig. 3B of the provided PDF file, the number of EV-A71-C1-16-infected leukocyte cells (CellTracker-positive cells; green) attached with BLECs seem not to be much different with those for MI and C1-06. But please ignore any misunderstanding on my interpretation.

Reply. We agree with the reviewer that the images are not representative of the data. We have chosen a more appropriate image for EV-A71-C1-16-infected leukocyte cells in Fig. 3B.

4. Fig. 3D; For the comparison, viral RNA from the upper compartment should be also assessed.

Reply. We thank the reviewer for this comment. For technical reasons, we did not perform FACS analyses on the samples collected from the upper compartment during our experiments. However, we determined a crude viral load for the whole cells present within the upper compartment after the 17-hour-transmigration assay. We have reported the data in the new figure S8 and described the results in lines 202 – 205 of the revised manuscript *"In the upper compartment, the amounts of viral RNAs were considerably higher than those detected in the lower compartment (Fig. S8), demonstrating again the role of the BBB in impeding the passage of viral RNA or virus particles."*

5. Page 7, lines 185-188; for immune cell transmigration assay, only two EV-A71 strains (C1 and C1-like) were used instead of three strains (Fig. 3).

Reply. Correct. For this investigation, we chose to restrict our comparison to the C1-06 and C1-16 virus strains because we found limited differences in the levels of viral RNA produced in leucocytes between C4 and C1-16. We have indicated this in lines 173 and 174.

Re: Spectrum00690-24R1 (Enterovirus A71 crosses a human blood-brain barrier model through infected immune cells)

Dear Dr. Jean-Luc Bailly:

Your manuscript has been accepted, and I am forwarding it to the ASM production staff for publication. Your paper will first be checked to make sure all elements meet the technical requirements. ASM staff will contact you if anything needs to be revised before copyediting and production can begin. Otherwise, you will be notified when your proofs are ready to be viewed.

Sincerely,
Peter Pelka
Editor
Microbiology Spectrum